# Assessment of the Possibility of Using the Laryngoscopes Macintosh, McCoy, Miller, Intubrite, VieScope and I-View for Intubation in Simulated Out-of-Hospital Conditions by People without Clinical Experience: A Randomized Crossover Manikin Study

**DOI:** 10.3390/healthcare11050661

**Published:** 2023-02-23

**Authors:** Paweł Ratajczyk, Przemysław Kluj, Przemysław Dolder, Bartosz Szmyd, Tomasz Gaszyński

**Affiliations:** 1Department of Anesthesiology and Intensive Care, Medical University of Lodz, 90-549 Lodz, Poland; 2Department of Pediatrics, Oncology and Hematology, Medical University of Lodz, 90-549 Lodz, Poland

**Keywords:** videolaryngoscopes, intubation, out-of-hospital settings

## Abstract

The aim of the study was to evaluate the laryngoscopes Macintosh, Miller, McCoy, Intubrite, VieScope and I-View in simulated out-of-hospital conditions when used by people without clinical experience, and to choose the one that, in the case of failure of the first intubation (FI), gives the highest probability of successful second (SI) or third (TI). For FI, the highest success rate (HSR) was observed for I-View and the lowest (LSR) for Macintosh (90% vs. 60%; *p* < 0.001); for SI, HSR was observed for I-View and LSR for Miller (95% vs. 66,7%; *p* < 0001); and for TI, HSR was observed for I-View and LSR for Miller, McCoy and VieScope (98.33% vs. 70%; *p* < 0.001). A significant shortening of intubation time between FI and TI was observed for Macintosh (38.95 (IQR: 30.1–47.025) vs. 32.4 (IQR: 29–39.175), *p* = 0.0132), McCoy (39.3 (IQR: 31.1–48.15) vs. 28.75 (IQR: 26.475–35.7), *p* < 0.001), Intubrite (26.4 (IQR: 21.4–32.3) vs. 20.7 (IQR: 18.3–24.45), *p* < 0.001), and I-View (21 (IQR: 17.375–25.1) vs. 18 (IQR: 15.95–20.5), *p* < 0.001). According to the respondents, the easiest laryngo- scopes to use were I-View and Intubrite, while the most difficult was Miller. The study shows that I-View and Intubrite are the most useful devices, combining high efficiency with a statistically significant reduction in time between successive attempts.

## 1. Introduction

Ensuring airway patency is the primary task of a paramedic in a patient with symptoms of respiratory failure [1]. It enables the delivery of oxygen to the lungs and the elimination of carbon dioxide from the body [2]. Various devices are used to obtain airway patency, e.g., oropharyngeal, nasopharyngeal, or supralaryngeal airway devices. However, the gold standard to ensure airway patency and at the same time to protect the lungs against the aspiration of food content is endotracheal intubation [2]. Correct intubation requires not only theoretical knowledge but also considerable manual skills, which deteriorate if not constantly improved [3]. This especially applies to people who do not perform it on a daily basis [1].

In out-of-hospital conditions, endotracheal intubation is most often performed at the ground level in conditions requiring the adoption of non-physiological and non-ergonomic body positions, often in unfavorable environmental conditions. This results in a significantly reduced level of comfort for the professional, which together with the stressful situation related to the patient’s life-threatening condition and responsibility for his or her health may translate into the effectiveness of intubation [4].

Difficult or failed tracheal intubation is a well-known cause of morbidity and mortality associated with anesthesia and emergency medicine [5]. It has been proven that repeated intubation attempts are associated with an increased incidence of adverse events [6], transport delay, prolonged hospitalization, poorer neurological outcomes [7] and increased mortality [8]. In the hospital setting, video laryngoscopy has been shown to reduce the number of failed intubations, improve the view of the glottis, and reduce airway trauma [1]. However, there are only a few heterogeneous studies comparing video laryngoscopy and direct laryngoscopy in the pre-hospital setting [9]. Moreover, in pre-hospital care, the success of intubation depends not only on the type of laryngoscope used, but also on the training and experience of the healthcare provider with the device.

All these factors result in prolonged intubation, when intubation in out-of-hospital conditions are performed by people with little experience [10].

Therefore, it seems reasonable to search for a device whose use by people with little or minimal clinical experience will result in the most effective and quickest endotracheal intubation, and at the same time will result in the shortest learning effect in the event of potential failures [4].

The aim of the study was to assess the possibility of using the following laryngoscopes, Macintosh, Miller, McCoy, Intubrite, VieScope and the I-View video laryngoscope, in simulated out-of-hospital conditions by providers without clinical experience, and to choose the laryngoscope among them that, in the case of a failed first intubation, offers the greatest possibility of successful second or third intubation as soon as possible. The secondary aim was to assess the learning and teaching aspect of laryngoscopy for paramedics regarding the third attempt of intubation using videodevices or other laryngoscopes.

In the available literature, there are little data comparing intubation times in consecutive intubation attempts. It seems to us that there is quite a significant dependency conditioning the potential usefulness of a given device in medical rescue, especially when it is used by people without clinical experience, as repeated, prolonged intubation attempts are associated with a later poor prognosis in patients [7].

## 2. Materials and Methods

### 2.1. Materials

In the study, we compared the majority of laryngoscopes available on the market that enable direct laryngoscopy, Macintosh (HEINE Optotechnik GmbH & Co. KG, Gilching, Germany), Miller (Scope Medical Devices Pvt. Ltd., Ambala City, India), McCoy (McCoy Truphatek, Jerusalem, Israel), Intubrite^®^ (LLC; Vista, CA, USA), VieScope^®^ (Adroit Surgical, Oklahoma City, OK, USA) with a dedicated 15 Fr Voir Bougie guidewire, and I-View™ VL video laryngoscope (Intersurgical Ltd., Wokingham, Berkshire, UK), in a simulated out-of-hospital setting when used by people with little clinical experience on a manikin model (Laerdal Airway Management Trainer Stavanger Norway manikin of universal difficulty) (Figure 1.).

Endotracheal tubes No. 7 were used for intubation. In each case, the endotracheal tubes and guides were covered with a standard lubricant dedicated to simulators. Simulated out-of-hospital conditions were created by placing the manikin in a neutral position at floor level.

### 2.2. Study Design

The study was conducted from 21 February 2021 to 8 June 2021 at the Norbert Barlicki University Teaching Hospital No. 1 in Lodz. Sixty randomly selected students in the third year of Paramedic Science, full-time first-cycle studies at the Medical University of Lodz, qualified for the study. All students signed informed consent for voluntary participation in the study.

The exclusion criterion was prior clinical experience with the laryngoscopes used in the study.

All participants listened to a 45 min lecture on the construction of laryngoscopes and the principles of using them, as well as the anatomical structure and the method and technique of intubation. After the presentation, the instructor presented the correct intubation with each of the 6 tested laryngoscopes. Then, under the supervision of the teacher, the students participated in the workshop where they had the opportunity to intubate a manikin placed on the operating table at the optimal height for each participant with each of the tested laryngoscopes. After a month, 60 students took part in the actual study.

### 2.3. Study Protocol

After signing their informed voluntary consent to participate in the study, the following demographic and medical data of the test participants were recorded in pseudonymized form:SexAgeExperience level: the number of dummy intubations performed so far by the subject and which laryngoscopes were used for previous intubations.

Participants were asked to perform three endotracheal intubations on a certified airway training manikin (Laerdal Airway Management Trainer Stavanger Norway, universal difficulty) placed at floor level in a neutral position (out-of-hospital simulation), using each of the evaluated laryngoscopes.

Each participant used all devices in random order in a crossover arrangement. The order in which the laryngoscopes were used was randomized using sealed opaque envelopes. The locked randomization strategy was generated using the Randomizer Program (randomizer.org). Flow diagram is presented in Figure 1.

Timing began with taking the laryngoscope and ended with initial ventilation with a resuscitation bag after placement and sealing of the endotracheal tube. Intubation was considered successful after confirming the breathing movements of the manikin’s lungs. The attempt was defined as a failure in the absence of manikin breathing movements or for an intubation time of more than 60 s. The criterion of over 60 s defining the intubation attempt as unsuccessful was adopted due to the fact that the study was to assess the usefulness of the devices by people without clinical experience in intubation.

After each intubation attempt with a given laryngoscope, two subsequent intubation attempts with the same device were made. After the completion of three intubations with a given laryngoscope, there was a break of at least 2 h (in order to eliminate the impact of intubation with a given laryngoscope on the use of the next device). After the break, the subject proceeded to three intubations of the manikin with a randomly selected device.

The subject assessed intubation with a given laryngoscope on the basis of a subjective assessment of tracheal intubation difficulty (number rating scale 0–10, 0: no difficulty, 10: highest difficulty).

The following data were pseudonymously recorded for all simulations:Success of intubation, position of the tube: tracheal vs. esophageal (primary endpoint);Comparison of times to ventilation in the first, second, and third intubation attempts (secondary endpoint);Feelings of subjects (secondary endpoint).

### 2.4. Statistical Analysis

The distribution of continuous data was checked with the Shapiro–Wilk test. As the average time of intubation has a distribution other than normal for at least one laryngoscope (*p* < 0.05), continuous data were presented as median with IQR. Furthermore, the dependencies between them were assessed with the Kruskal–Wallis test with Dunn’s post hoc tests. Dependencies for dependent data (comparisons between approaches) were assessed with the usage of the t-student test for dependent data in the case of normal distribution and Wilcoxon’s test in other cases. In both cases, the Bonferroni correction was used. Nominal data were present as *n* (% of total) and assessed with a test chosen based on the size of the smallest subgroup. The statistical analysis was performed using Statistica 13.1PL (StatSoft, Poland, Krakow).

## 3. Results

### 3.1. Demographic and Contextual Data

The study included 60 third-year students of Paramedic Science (18 women and 42 men). The average age of the respondents was 22 years. Among the surveyed, 21 students had intubated the manikin fewer than 10 times so far, 22 students had performed between 10 and 20 only manikin intubations so far, and 17 students had performed more than 20 only manikin intubations. Before, everyone had used only the Macintosh laryngoscope for only manikin intubation.

### 3.2. Primary Endpoint

For the first intubation, the highest success rate was observed for the I-View laryngoscope and the lowest for the Macintosh laryngoscope: 54 (90%) vs. 36 (60%; *p* < 0.001). In the case of the second intubation, the highest success rate was observed for the I-View laryngoscope and the lowest for the Miller laryngoscope: 57 (95%) vs. 40 (66.7%; *p* < 0.001). In the case of the third intubation, the highest success rate was again observed for the I-View laryngoscope, and the lowest this time for the Miller laryngoscope, McCoy laryngoscope and VieScope laryngoscope: 59 (98.33%) vs. 42 (70%; *p* < 0.001; see Table 1).

There were no significant dependencies in the success rate between first and second attempts, second and third attempts, and first and third attempts (see Figure 2). Comparing all laryngoscopes, the highest intubation efficiency was obtained for the I-View laryngoscope (90%, 95%, 98.33%), followed by the Intubrite laryngoscope (83.33, 88.3%, 91.67%) and the VieScope laryngoscope (65%, 80%, 70%). The effectiveness of the remaining laryngoscopes, Macintosh, McCoy and Miller, oscillated between 60% and 73.33% (see Table 1). An increasing learning curve in the use of the tested laryngoscopes was observed only for laryngoscopes I-View and Intubrite (see Figure 2).

### 3.3. Secondary Endpoints

There were significant differences between the mean time of intubation with the usage of the aforementioned laryngoscopes (*p* < 0.001). The statistically significant results of the performed post hoc Dunn’s test are shown in Figure 3.

A significant shortening of intubation time between the first and the third intubation was observed for the Macintosh laryngoscope (38.95 (IQR: 30.1–47.025) vs. 32.4 (IQR: 29–39.175), *p* = 0.0132), McCoy laryngoscope (39.3 (IQR: 31.1–48.15) vs. 28.75 (IQR: 26.475–35.7), *p* < 0.001), Intubrite laryngoscope (26.4 (IQR: 21.4–32.3) vs. 20.7 (IQR: 18.3–24.45), *p* < 0.001), and I-View laryngoscope (21 (IQR: 17.375–25.1) vs. 18 (IQR: 15.95–20.5), *p* < 0.001). Additionally, a significant shortening of intubation time between the first vs. second attempt and the second vs. third attempt was observed only for Intubrite and I-View laryngo scopes. In the case of the McCoy laryngoscope, a significant improvement was observed between the second and third approaches and the first and third approaches (see Figure 4).

According to the respondents, the easiest laryngoscope to use was the I-View laryngoscope, then the Intubrite, Macintosh, and McCoy, and finally the two laryngoscopes with straight blades: Miller and VieScope (see Figure 5).

## 4. Discussion

A significant reduction in intubation time between the first and third intubations was observed for the Macintosh laryngoscope, the McCoy, Intubrite laryngoscope and I-View laryngoscope. In addition, a significant reduction in intubation time between the first and second attempts and the second and third attempts was observed only with the Intubrite and I-View laryngoscopes. For the McCoy laryngoscope, there was a significant improvement in intubation times between the second and third attempts and the first and third attempts.

The I-View laryngoscope turned out to be the easiest device to use in relation to the feelings of the subjects. This is probably due to the fact that there is no need to keep a straight line between the eyes of the professional and the glottis. In simulation, where the manikin was intubated at the floor level, the lack of the need to maintain this line is important because it does not require the intubating person to assume a more forced, bent body position, which is uncomfortable and non-ergonomic [3]. In the case of the I-View laryngoscope, the possibility of evaluating the view of the glottis thanks to the device’s monitor makes the assumed body position less bent and more friendly to the examined person [3]. This is essential when a patient is intubated by people without experience in airway management. In this situation, if there is a choice between a Macintosh laryngoscope and video laryngoscopes, including I-View, some authors suggest choosing the latter [11].

In the case of intubation by anesthesiologists, Wakabayashi believes that despite the fact that video laryngoscopes give better visibility of the glottis and are easier to use, the effectiveness and times of intubation with a classic Macintosh laryngoscope are at an acceptable level. This is vital given the widespread availability of Macintosh laryngoscopes and the still limited availability of video laryngoscopes [12].

Among the video laryngoscopes, some authors suggest that the I-View laryngoscope is a suitable device for use in difficult conditions of pre-hospital care due to its ease and single use [13]. In their study, Maritz et al. showed that the use of video laryngoscopy provided better intubation conditions, enabled better visualization of the glottis, and thus facilitated intubation when used not only by anesthesiologists with extensive experience in conventional and video laryngoscopy, but also paramedics with little previous experience in conventional and non-conventional experience in video laryngoscopy [10,14]. Although the use of video laryngoscopes did not affect the success of intubation among anesthesiologists, in the hands of paramedics with little experience in intubation it reduced the failure rate from 14.8% for the conventional Macintosh laryngoscope to 3.7% for the video laryngoscope [10].

The high position of the Intubrite laryngoscope is probably related to the new, ergonomic handle of this laryngoscope [3]. The introduction of more ergonomic devices would reduce the professional’s workload, which is an important factor determining patient safety [5,15,16,17]. This applies in particular to people with little experience in intubation, in whom potential intubation difficulties may occur more often, especially in the group of obese patients. These patients, due to their physique and anatomy of the airways, may require greater strength to open the airways [18]. According to J. Tesler and J. Rucker, when the Intubrite laryngoscope is used in out-of-hospital conditions the percentage of the need for repeated intubation attempts and the percentage of tooth damage decreased compared to the Macintosh laryngoscope [4]. Similar results were obtained by T. Gaszyński, who stated that in the case of the Intubrite laryngoscope the patient’s body is less traumatized compared to Macintosh laryngoscope [19].

Macintosh and McCoy laryngoscopes in our study had similar first intubation success rates of 60% and 65%, respectively, second intubation success rates of 73.3%, and third intubation success rates of 73.3% and 70%, respectively.

Furthermore, both laryngoscopes showed a significant improvement in intubation time between the first and third attempts. Moreover, McCoy laryngoscope enabled improvement between the second and third attempts. Therefore, in the case of failure of the first intubation, they give a chance for the correct placement of the endotracheal tube by people without clinical experience in subsequent attempts. However, in terms of average intubation times, both laryngoscopes were inferior to the I-View and Intubrite laryngoscopes, yet the Macintosh laryngoscope turned out to be easier to use in our study.

There are different opinions in the literature regarding clinical situations in which one of these two laryngoscopes is more useful than the other.

In a similar research model in which inexperienced medical students intubated manikins with Macintosh and McCoy laryngoscopes, Higashizawa found that the time needed to correctly position the endotracheal tube was similar with both laryngoscopes but the McCoy laryngoscope was more difficult to operate. The author suggested that the Macintosh laryngoscope is more useful for teaching inexperienced medical students [18], whereas Yildirim showed that the use of the McCoy laryngoscope shortens and provides easier intubation than the use of the Macintosh laryngoscope [20]. However, Sethuraman came to different conclusions, stating that there is no advantage in using the McCoy laryngoscope over the Macintosh laryngoscope in the examination on manikins with difficult airways [21]. In turn, in patients with limited mobility of the cervical spine, Uchida showed that the McCoy laryngoscope facilitates intubation compared to the Macintosh laryngoscope [22] and it is also superior to some videolaryngoscopes [23]. Similar conclusions were drawn by Gabbott and Maharaj [24,25]. However, the latter author believes that, although the McCoy laryngoscope improves the visualization of the larynx more than the Macintosh laryngoscope in patients with both normal and difficult airways, reducing the number of intubation attempts and the number of optimization maneuvers required, it has proven to be more difficult and less reliable than the Macintosh laryngoscope [25,26,27,28,29,30,31]. In patients with morbid obesity, Nandakumar et al. found the McCoy laryngoscope to be as effective as the Macintosh laryngoscope, and concluded that due to its widespread availability and familiarity the latter laryngoscope should be used in this group of patients [26].

In our study, the successful first, second, and third intubation rates with the Miller laryngoscope were 73.3%, 66.7%, and 70%, respectively. There was no statistically significant reduction in intubation time between successive intubation attempts. It also turned out to be the most difficult laryngoscope to use among our subjects. Such a distant position of this laryngoscope in our list is probably due to the fact that the need to maintain a straight line between the subject’s eye and the entrance to the airway in the case of intubation of a manikin lying at the floor level requires adopting the least comfortable position of the body. The lack of or little possibility of lifting the epiglottis when using this laryngoscope also affects the effort of the professional. Vidhya came to different conclusions, believing that the Miller’s laryngoscope enables much better visualization of the larynx than the McCoy and Macintosh laryngoscope, even in patients with difficult airways [31]. Similarly, Achen claimed that Miller’s laryngoscope enabled better visualization of the airway entrance than the Macintosh laryngoscope, and therefore everyone should learn laryngoscopy using both laryngoscopes [32]. This is important because, according to other authors, although the view of the glottis was better with the Miller laryngoscope than with the Macintosh laryngoscope, intubation conditions turned out to be better with the Macintosh laryngoscope [33,34]. The Miller laryngoscope was superior to the Macintosh and McCoy laryngoscope for visualizing the glottis in children [35,36].

The VieScope laryngoscope, a variant of the Miller laryngoscope requiring two-stage intubation, was found to be similarly effective during the first intubation as the McCoy and Macintosh laryngoscopes: 65%, 65%, and 60%, respectively. For the second intubation, its effectiveness increased to 80% and approached that of the Intubrite laryngoscope (88.3%), while during the third intubation, its effectiveness decreased to 70%. There was no statistically significant difference between intubation times in consecutive trials. According to the respondents, this device was also as difficult to use as the Miller laryngoscope. Such a low rank of this laryngoscope, and likewise the Miller laryngoscope, may result from the need to maintain the line of the intubating eye to the entrance to the airway and the need to adopt a more strenuous body position compared to the I-View, Intubrite, McCoy, and Macintosh laryngoscopes. The VieScope laryngoscope was originally designed for battlefield medicine, to facilitate the intubation of patients with difficult airways by being always ready for use and by focusing light on target tissues. This was confirmed in Maślanka’s study, which showed that, taking difficult airways into consideration, the VieScope laryngoscope compared to the Macintosh laryngoscope had a shorter intubation time and a higher success rate on the first attempt [37]. Similar conclusions were drawn by Wieczorek et al., who compared the use of bébé VieScope and direct laryngoscopy during emergency intubation on a model of a pediatric manikin performed by paramedics with and without personal protective equipment [38]. In their prospective, multicenter, randomized study, Szarpak et al. proved that the VieScope laryngoscope enables more effective and faster intubation than the Macintosh laryngoscope in patients with suspected or confirmed diagnosis of COVID-19, who required pre-hospital cardiopulmonary resuscitation. In these studies, the study group consisted of paramedics with clinical experience and the ability to use various laryngoscopes. In our case, there was no scenario imitating difficult airways, which could result in the lack of advantage of this laryngoscope over other devices [39]. Additionally, the study group consisted of people without clinical experience. Another difficulty for the participants in the study was the fact that it requires two stages to intubate, which can make it difficult for inexperienced people to use. This translated into a result similar to that of the Miller laryngoscope in terms of reported subjective intubation difficulties.

Similar conclusions were reached by Ecker et al., who conducted their study on a manikin under simulated conditions of massive regurgitation. In the case of patients with lower esophageal sphincter insufficiency, intubation with the VieScope laryngoscope compared to the Macintosh laryngoscope turned out to be longer, similar to our study, and resulted in a greater amount of aspirated content into the airways. The study group consisted of experienced anesthesiologists, i.e., people who perform intubation on a daily basis and have experience in solving various situations that may occur during intubation [40].

The longer intubation time of the VieScope laryngoscope compared to other airway devices was again noted by Ecker when he compared it to the Glidescope video laryngoscope in both simulated normal and difficult airways [41]. The prolongation of intubation time using the VieScope laryngoscope was also found in the case of intubation of patients qualified for elective surgical procedures, with no advantage of this laryngoscope over the Macintosh laryngoscope in this group of patients [42].

The study showed that it is necessary to constantly practice methods of airway management, including endotracheal intubation [27,28,29]. It is particularly important to learn how to use multiple laryngoscopes, as it may be useful in unconventional situations requiring the modification of technique, equipment or body position [33]. Each exercise in this area reduces the risk of making a mistake, reduces the stress of people performing a given procedure and, most importantly, increases the chance of survival of the patient and their return to the state before the event [33]. A similar conclusion was drawn by Pieters et al. from their study comparing seven videolaryngoscopes in manikin settings [42]. They compared the Macintosh classic laryngoscope, Airtraq, Storz C-MAC, Coopdech VLP-100, Storz C-MAC D-Blade, GlideScope Cobalt, McGrath Series5, and Pentax AWS. They observed 65 anesthetists, 67 residents in anesthesia, 56 paramedics and 65 medical students, intubating the trachea of a standardized manikin model. The results underline the importance of variability in device performance across individuals and staff groups, which has important implications for which devices hospital providers should rationally use. It is proven that videolaryngoscopes offer a better view of the entrance to larynx [43], and therefore reduce the risk of possible injuries related to intubation efforts [44]; however, training is still needed to avoid possible problems with the use of videolaryngoscopy [45,46]. Using these tools for learning purposes for unexperienced providers, in addition, may provide greater applicability [43,47,48].

The study has several limitations. Firstly, it was conducted on a manikin model, where simulated out-of-hospital conditions were created by placing the manikin at floor level, without the influence of other external factors affecting the effectiveness of intubation. Secondly, difficult airway scenarios were not also studied. Finally, the study group consisted of Paramedic Science students who, nevertheless, had little previous experience in intubating a dummy with a Macintosh laryngoscope due to their limited years of study.

## 5. Conclusions

Taking into account the results of the study, the I-View and Intubrite laryngoscopes turned out to be the most useful devices for intubation in simulated out-of-hospital conditions by people with no clinical experience. They combined high efficiency of intubation with statistically significant shortening of intubation times between successive attempts. Due to the small study group and the manikin model, additional studies should be conducted on a larger group of subjects.

## Data Availability

Data available on request due to restrictions eg privacy or ethical.

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
