# Peer review of "Assessment of the Possibility of Using the Laryngoscopes Macintosh, McCoy, Miller, Intubrite, VieScope and I-View for Intubation in Simulated Out-of-Hospital Conditions by People without Clinical Experience: A Randomized Crossover Manikin Study"

_healthcare, 2023, doi:10.3390/healthcare11050661_

Round 1
Reviewer 1 Report
Whether the study was registered.
Has a sample size calculation been performed?
Authors indicate that "The exclusion criterion was prior clinical experience with the laryngoscopes used in 140 the study" - did it refer to simulation or clinical use? – this is in opposition to section 3.1.
Please don't use therm "dummy" but "simulation"
Figure 1 needs improvement - the signature is also missing.
The authors should maintain standardization - in one graph they give data as a median and in the second as an average - example Figure 2 and 3.
Authors should - since VieScope is a new method - also include the following publications:
Wieczorek P, Szarpak L, Dabrowska A, Pruc M, Navolokina A, Raczynski A, Smereka J. A Comparison of the bébé VieScope™ and Direct Laryngoscope for Use While Wearing PPE-AGP: A Randomized Crossover Simulation Trial. Children (Basel). 2022 Nov 18;9(11):1774. doi: 10.3390/children9111774.
Szarpak L, Peacock FW, Rafique Z, Ladny JR, Nadolny K, Malysz M, Dabrowski M, Chirico F, Smereka J. Comparison of Vie Scope® and Macintosh laryngoscopes for intubation during resuscitation by paramedics wearing personal protective equipment. Am J Emerg Med. 2022 Mar;53:122-126. doi: 10.1016/j.ajem.2021.12.069.
As well as conduct discussions in relation to a larger number of publications in the field of the analyzed methods of intubation, including:
doi: 10.7759/cureus.24914.
doi: 10.4103/aer.aer_163_21.
doi: 10.1016/j.ajem.2015.09.005.
doi: 10.1007/s11739-016-1437-3.
doi: 10.1097/SIH.0000000000000161.
doi: 10.1016/j.ajem.2018.01.081.
doi: 10.1016/j.ajem.2017.03.006.
doi: 10.1016/j.ajem.2017.03.067.
doi: 10.1016/j.ajem.2017.01.005.
doi: 10.1007/s00431-015-2567-8.
Author Response
Comments and Suggestions for Authors
Whether the study was registered.
Answer: The study was not registered because it was performed on manikins
Has a sample size calculation been performed?
Answer: We did not perform a sample size calculation. In the current study, we have examined all 3rd-year paramedic trainees studying at the Medical University of Lodz. This approach facilitates obtaining a homogeneous group. Based on similar published studies, we considered the study group to be sufficient for example: Wieczorek P, Szarpak L, Dabrowska A, Pruc M, Navolokina A, Raczynski A, Smereka J. A Comparison of the bébé VieScope™ and Direct Laryngoscope for Use While Wearing PPE-AGP: A Randomized Crossover Simulation Trial. Children (Basel). 2022 Nov 18;9(11):1774. doi: 10.3390/children9111774.
Authors indicate that "The exclusion criterion was prior clinical experience with the laryngoscopes used in 140 the study" - did it refer to simulation or clinical use? – this is in opposition to section 3.1.
Answer: It refer to clinical use.
Line 140 it was corrected from: The exclusion criterion was prior clinical experience with the laryngoscopes used in the study, to: The exclusion criterion was only prior clinical experience with the laryngoscopes used in the study (P3 Line 91-92).
Section 3.1 details that the students' previous experience was only with manikin intubation. Line 210 it was corrected from: between 10 and 20 intubations so far, and 17 students have performed more than 20 intubations, to: between 10 and 20 only manikin intubations so far, and 17 students have performed more than 20 only manikin intubations (P 6, Line 158-159).
Line 211 it was corrected from: Before everyone had used only the Macintosh laryngoscope for intubation, to: Before everyone had used only the Macintosh laryngoscope for only manikin intubation (P 6 Line 159-160).
Please don't use therm "dummy" but "simulation"
Answer: Corrected
Line 299 and 300 it was corrected from: In simulated out-of-hospital conditions, where the dummy was intubated at the floor level,…to: In simulation, where the manikin was intubated at the floor level,…( P 9, Line 215-216)
Figure 1 needs improvement - the signature is also missing.
Answer: Figure 1 has been corrected. The signature is placed on the line 143 and 144. The earlier signature Fig 1 has been removed from the line 191-192
The authors should maintain standardization - in one graph they give data as a median and in the second as an average - example Figure 2 and 3.
Answer: An error occurred while translating the text into English. The word “median” has been inserted in place of “mean". Line 235 it was corrected from: There were significant differences between the median time of intubation, to: There were significant differences between the mean time of intubation (P 4, Line 184).Line 238 it was corrected from: The median time of intubation in different intubation approaches, to The mean time of intubation in different intubation approaches ( P 7, Line 188 ).Line 251 it was corrected from: Graph of average intubation times with a given laryngoscope…,to: Graph of mean intubation times with a given laryngoscope….(P 8, Line 199-200)
Authors should - since VieScope is a new method - also include the following publications:
Wieczorek P, Szarpak L, Dabrowska A, Pruc M, Navolokina A, Raczynski A, Smereka J. A Comparison of the bébé VieScope™ and Direct Laryngoscope for Use While Wearing PPE-AGP: A Randomized Crossover Simulation Trial. Children (Basel). 2022 Nov 18;9(11):1774. doi: 10.3390/children9111774.
Answer: the publication was used in the discussion P 11 line 319 - 322. Reference 38
Similar conclusions were drawn by Wieczorek at al, who compared the use of bébé VieScope and direct laryngoscopy during emergency intubation on a model of pediatric manikin performed by paramedics with and without personal protective equipment –
Szarpak L, Peacock FW, Rafique Z, Ladny JR, Nadolny K, Malysz M, Dabrowski M, Chirico F, Smereka J. Comparison of Vie Scope® and Macintosh laryngoscopes for intubation during resuscitation by paramedics wearing personal protective equipment. Am J Emerg Med. 2022 Mar;53:122-126. doi: 10.1016/j.ajem.2021.12.069.
Answer: the publication was used in the discussion P 11, line 322 - 329, reference 39
In their prospective, multicenter, randomized study, Szarpak and al proved that the VieScope laryngoscope enables more effective and faster intubation than the Macintosh laryngoscope in patients with suspected or confirmed diagnosis of COVID-19, who required pre-hospital cardiopulmonary resuscitation
As well as conduct discussions in relation to a larger number of publications in the field of the analyzed methods of intubation, including:
doi: 10.7759/cureus.24914.
Answer: This publication is about the hemodynamic response to intubation. It is not studied in our study
doi: 10.4103/aer.aer_163_21.
Answer: This publication is already used in the discussion, reference 31
doi: 10.1016/j.ajem.2015.09.005.
Answer: the publication was used in the discussion, reference 14.
doi: 10.1007/s11739-016-1437-3.
Answer: This publication is already used in the discussion, reference 20.
doi: 10.1097/SIH.0000000000000161.
Answer: the publication was used in the discussion, reference 23.
doi: 10.1016/j.ajem.2018.01.081.
Answer: the publication was used in the discussion, reference 15.
doi: 10.1016/j.ajem.2017.03.006.
Answer: the publication was used in the discussion, reference 16.
doi: 10.1016/j.ajem.2017.03.067.
Answer: the publication was used in the discussion, reference 48.
doi: 10.1016/j.ajem.2017.01.005.
Answer: the publication was used in the discussion, reference 17.
doi: 10.1007/s00431-015-2567-8.
Answer: this test is performed on a pediatric manikin. In our study, we used a standard manikin
Reviewer 2 Report
The introduction section is too long. Make the introduction section half as long.
・Please delete P3 L108-129.
・P4 Please delete L171-176 (below Subject~).
・Figure 1 is difficult to understand. Please correct.
・Describe and clarify all statistical results for the primary endpoint. (including results with no significant difference)
The discussion section is too long. Cut the discussion section in half.
・Please delete P9 L258-281.
・Please delete P9 L289-295.
Author Response
Comments and Suggestions for Authors
The introduction section is too long. Make the introduction section half as long.
Answer: Introduction has been shortened from page 1 the following lines have been removed: 32 – 44from page 2 the following lines have been removed: 50 – 61, 74-77.
・Please delete P3 L108-129.
Answer: lines 108 - 129 have been removed
・P4 Please delete L171-176 (below Subject~).
Answer: lines 171 – 176 (below Subject~) have been removed
・Figure 1 is difficult to understand. Please correct.
Answer: Figure 1 has been corrected.
・Describe and clarify all statistical results for the primary endpoint. (including results with no significant difference)
Answer: supplemented and clarified the results for the primary endpoint with the following content
P 6, line 173 – 178: Comparing all laryngoscopes, the highest intubation efficiency was obtained for the I-View laryngoscope (90%, 95%, 98,33%), followed by the Intubrite laryngoscope (83,33, 88,3%, 91,67%) and the VieScope laryngoscope (65%, 80%, 70%). The effectiveness of the remaining laryngoscopes: Macintosh, McCoy, Miller oscillated between 60% and 73,33% (see Table 1). An increasing learning curve in the use of the tested laryngoscopes was observed only for laryngoscopes: I-View and Intubrite (see Figure 2).
The discussion section is too long. Cut the discussion section in half.
Answer: Corrected
from page 10 the following lines have been removed: 325 – 337, 348 - 350from page 11 the following lines have been removed: 381 – 393
Due to the suggestions of the first reviewer and his recommendations regarding the need to extend the discussion to further references, I cannot shorten the discussion any further.
・Please delete P9 L258-281.
Answer: lines 258 – 281 have been removed.
・Please delete P9 L289-295.
Answer: lines 289 – 295 have been removed
Reviewer 3 Report
Congratulations. This is a nice well-written study on an important subject that might contribute to save lives. For that, it deserves all my support.
I think that would be of great benefit to do the same study in different conditions:
1- If some prehospital system would choose to adopt one of those devices, it is expected that some (if not most) of the paramedics have some previous experience with intubation. How would they perform with this change? So, I think you should include paramedics with experience in your study to understand the possible impact of such change.
2- Real scenario tests are of paramount importance. I realise their difficulty but any conclusions based on mannequin tests only could be invalid.
Furthermore, costs are important when you have to make decisions. So, at least, the costs of each one of these different solutions must be known.
So, I would suggest you to include experienced paramedics in the study and to inform about the prices of the different equipment.
A future study in real scenario situations should be planned.
Author Response
Comments and Suggestions for Authors
Congratulations. This is a nice well-written study on an important subject that might contribute to save lives. For that, it deserves all my support.
I think that would be of great benefit to do the same study in different conditions:
1- If some prehospital system would choose to adopt one of those devices, it is expected that some (if not most) of the paramedics have some previous experience with intubation. How would they perform with this change? So, I think you should include paramedics with experience in your study to understand the possible impact of such change.
2- Real scenario tests are of paramount importance. I realise their difficulty but any conclusions based on mannequin tests only could be invalid.
Furthermore, costs are important when you have to make decisions. So, at least, the costs of each one of these different solutions must be known.
So, I would suggest you to include experienced paramedics in the study and to inform about the prices of the different equipment.
A future study in real scenario situations should be planned.
Answer: thank you very much for the review. All comments will be taken into account when planning further research work